# Crykey: Rapid identification of SARS-CoV-2 cryptic mutations in wastewater

Yunxi Liu [1], Nicolae Sapoval[1], Pilar Gallego-García [2,3], Laura Tomás [2,3], David Posada [2,3,4], Todd J. Treangen [1] ✉ & Lauren B. Stadler [5] ✉

Wastewater surveillance for SARS-CoV-2 provides early warnings of emerging variants of concerns and can be used to screen for novel cryptic linked-read mutations, which are co-occurring single nucleotide mutations that are rare, or entirely missing, in existing SARS-CoV-2 databases. While previous approaches have focused on specific regions of the SARS-CoV-2 genome, there is a need for computational tools capable of efficiently tracking cryptic mutations across the entire genome and investigating their potential origin. We present Crykey, a tool for rapidly identifying rare linked-read mutations across the genome of SARS-CoV-2. We evaluated the utility of Crykey on over 3,000 wastewater and over 22,000 clinical samples; our findings are three-fold: i) we identify hundreds of cryptic mutations that cover the entire SARS-CoV-2 genome, ii) we track the presence of these cryptic mutations across multiple wastewater treatment plants and over three years of sampling in Houston, and iii) we find a handful of cryptic mutations in wastewater mirror cryptic mutations in clinical samples and investigate their potential to represent real cryptic lineages. In summary, Crykey enables large-scale detection of cryptic mutations in wastewater that represent potential circulating cryptic lineages, serving as a new computational tool for wastewater surveillance of SARS-CoV-2.

Wastewater monitoring is a vital tool complementing clinical testing for COVID-19 surveillance[1–9] and can fill in the surveillance gap when clinical testing is unavailable or halted. Multiple studies have demonstrated that SARS-CoV-2 variants of concern (VOCs) can be detected in wastewater samples[10–14], preceding clinical testing by up to 2 weeks[8]. Furthermore, wastewater samples contain information on the genomic diversity of the circulating variants in the entire community, avoiding the sampling bias inherent to the clinical surveillance, which focuses on symptomatic patients[15–17]. Importantly, wastewater monitoring can also detect novel and rare SARS-CoV-2 lineages not represented in GISAID's EpiCoV database[18], termed cryptic lineages[19]. A few methods have been proposed for the detection of cryptic lineages from wastewater samples. Still, they often require a combination of ultra-deep sequencing of specific genomic regions as well as a mixture of short-read, long-read, and proximity ligation sequencing technologies and

thus are not compatible with most wastewater sequencing protocols used for routine monitoring due to time and cost limitations[15,20]. Moreover, non-uniform sequencing coverage caused by amplicon efficiency heterogeneity and environmental RNA degradation creates a challenge for detecting cryptic lineages from wastewater samples[4,21,22]. Furthermore, the origin of these cryptic lineages in wastewater is still an open question[23]. It has been proposed that they could be rare intra-host lineages not represented in the consensus genomes available in public databases, rare lineages with low prevalence in the population, lineages from non-human hosts (like rats), or technical artifacts[19,20,24].

The objective of this study was to (1) develop a tool that enables the detection in wastewater samples of cryptic lineages that have not, or rarely been reported in GISAID's EpiCoV database, (2) investigate the possible origins of these cryptic lineages by contextualizing wastewater and clinical surveillance data. In this manuscript, we introduce

[1]Department of Computer Science, Rice University, Houston, TX 77005, USA. [2]CINBIO, Universidade de Vigo, 36310 Vigo, Spain. [3]Galicia Sur Health Research Institute (IIS Galicia Sur), SERGAS-UVIGO, Vigo, Spain. [4]Department of Biochemistry, Genetics, and Immunology, Universidade de Vigo, 36310 Vigo, Spain. [5]Department of Civil and Environmental Engineering, Rice University, Houston, TX 77005, USA. ✉e-mail: treangen@rice.edu; lauren.stadler@rice.edu

Crykey, a novel computational method for detecting rare linked-read mutations from wastewater samples that exploit the co-occurrence of point mutations on the same sequencing read or read-pair (from now on, linked-read or LR mutations). The rationale is that LR mutations found in wastewater samples but nonexistent or at a very low prevalence (e.g., <0.0001) in public databases represent potential cryptic lineages (from now on, potential cryptic lineage will be denoted as CR); i.e., rare linked-read mutations supported by 5 or more reads which we claim are indicative of one of the following: real cryptic lineages, SARS-CoV-2 transcription variation due to subgenomic mRNAs[25], or systematic artifacts. We used Crykey to analyze 3175 wastewater samples collected in Houston, Texas, USA, from February 2021 to November 2022. Our results are threefold: (i) We discover numerous cryptic mutations spanning the whole SARS-CoV-2 genome, (ii) we monitor the occurrence of these cryptic mutations across numerous wastewater treatment plants (WWTPs), observing them over a period of 3 years in Houston, and (iii) we identify cryptic mutations in wastewater samples that reflect those in clinical samples, and explore the possibility of these representing actually hidden lineages.

## Results

To evaluate the utility and efficiency of Crykey, we applied it to SARS-CoV-2 amplicon sequencing data from 3175 wastewater samples collected from 39 wastewater treatment plants (WWTPs) in Houston between February 2021 and November 2022, as well as in 5060 short-read clinical samples collected within the Greater Houston between December 2021 and January 2022 (Supplementary Fig. 1), and nearly 9000 short-read clinical samples collected outside of Texas over the same 8-week time period (between 2021-12-06 and 2022-01-31; Supplementary Fig. 2). In addition, we examined over 7000 long-read clinical samples on a specific CR. We will now delve into the specific results from these data.

### Overview of Crykey and computational performance

Crykey is a computational tool designed to search for cryptic mutations from samples by performing fast variant queries to determine how rare a set of LR mutations is among millions of publicly available genomes. Genomes of the same lineage during a short period of time have more mutations in common. Therefore, we indexed the Crykey database by partitioning the genomes into bins by sample collection date and lineage (Fig. 1a). By pre-computing the prevalence rate of each mutation in each bin, Crykey is able to quickly reduce the search space from the entire database to a few hundreds of genomes for exact matching (Fig. 1b, c, d). We identified a total of 6,744 CR candidates in wastewater samples from Houston. More than 67.8% of the candidate CRs were found in 50 or fewer sequences in the GISAID EpiCoV database. Fully novel CRs (zero prevalence, meaning it has not been previously observed) constitute more than 32.8% of the data (Fig. 2a). We benchmarked the processing time of exact searches on a Linux machine with Intel Xeon Gold 6138 2.00GHz CPU. The process time increases as the rarity of the CR candidates decreases, as the result of the expansion of the search space (Fig. 2b).

### Genomic distribution of CRs in wastewater samples

After quality control, we identified 705 CRs in the wastewater samples. Figure 3a shows the location of the CRs along the reference SARS-CoV-2 genome, their mean allele frequency (AF) across wastewater samples, the prevalence in GISAID, and the number of weeks (not necessarily consecutive) detected in wastewater. 74.8% of the CRs had a mean AF less than 0.2, while 7.8% of the CRs were detected at consensus-level AF (mean AF >=0.5) (Fig. 3a). The occurrence of the CRs varied significantly, ranging from 1 to 33 weeks (size of the dot in Fig. 3a). Almost half were located in the S (20.1%) and N (28.4%) genes. Most of the genome regions are dominated by CRs that contain only nonsynonymous mutations, except for gene N (Fig. 3b).

### Emergence of CRs co-occurs with the spread of new variants

The emergence of the CRs coincided with the spread of new VOCs. For example, the number of CRs and viral load in wastewater increased significantly around July 2021 (Fig. 4a), corresponding to the Delta wave in Houston (Fig. 4b). Similar patterns were observed during the emergence of B.1.1.529 (Omicron) in December 2021, BA.2 (Omicron) in May 2022, and BA.5 (Omicron) in July 2022. Most CRs could be associated with one (77%) or more (17%) known PANGO lineages circulating at the time (Supplementary section 1[26,27]). We observed fewer CRs between April and August 2022, when the sequencing breadth and depth of coverage dropped due to primer dropouts in lower-quality sequencing runs (thinner bars in Fig. 4a). It is hard to untangle whether this effect was due to specific genomic features of the BA.2 and BA.5 variants or whether it was a consequence of the lower quality of the sequencing data during this period.

### Houston CRs display distinct patterns

More than 400 CRs found in Houston wastewater were completely novel, or can only be found in less than 50 genomes in GISAID (Fig. 5). The vast majority of the CRs did not persist for long, with over 85% found in less than 4 consecutive weeks. Short-duration CRs (less than 4 weeks) were generally found in only a few WWTPs and at low AFs (Fig. 5). Interestingly, some CRs were detected in multiple WWTPs across the city and persisted for 4 months or longer. The most persistent CR observed, which we named CR12, was detected in the wastewater for 33 weeks. CR12 contains mutations A29039T and G29049A, which cause K256* (stop codon) and R259Q amino acid changes, respectively, on the N gene. The mean AF of CR12 across WWTPs was generally low (<0.1), with a few exceptions. CR12's presence ramped up slowly in 1–7 WWTPs, peaked in late November 2021 when observed in 16 WWTPs (Fig. 6), and phased out in late February 2022 and remained undetected for 2 months (but notice that the sequencing coverage also dropped during this period), re-appearing for a short time around May 2022.

Many CRs exhibited perplexing patterns of allele frequency, duration, and clinical sample prevalence. The first occurrence of CR12 in Houston wastewater can be traced back to Aug 2, 2021, when Delta was dominant in the community. As Omicron emerged in December 2021, CR12 continued to be present in the wastewater and clinical samples (Figs. 4 and 6). To evaluate the possibility of CR12 being a technical artifact, we first explored whether this could be due to a read mapping error by using a different read mapper, Bowtie2[28]. The results were consistent with those obtained previously with BWA MEM[29]. To investigate whether the mutations comprising CR12 were due to systematic sequencing errors, for example primer dependent errors, we further examined 7,113 clinical samples sequenced with the PacBio HiFi system (Sequel II), including 2,458 samples collected from Texas, and 4,655 samples collected from other US regions between Nov 06, 2021 and Mar 21, 2022. Forty-five of those samples included reads that supported the presence of CR12, and 28 were from Texas (Supplementary Table 1). While observing the rare LR mutations comprising CR12 in PacBio data does not fully rule out the possibility of technical artifacts, it provides a basis for ruling out platform-specific errors.

### Investigating CRs in clinical samples

As Omicron became dominant in the community, several CRs specific to the VOC emerged and became more prevalent among hosts (Fig. 7). We explored the occurrence of 20 CRs with short-term or long-lasting patterns in 5060 clinical samples collected within the Greater Houston (between 2021-12-06 and 2022-01-31; Supplementary Fig. 2). 12 out of 20 CRs detected in the wastewater were seen in the clinical samples (Fig. 7), including CR3, CR5, CR8, and CR12 (Fig. 8 and Supplementary Figs. 3–5). Remarkably, for these CRs, the mean AF within the clinical samples was very low (<0.05), except for CR5 (Supplementary Fig. 4). CR2 was associated with Delta, while the remaining eleven were

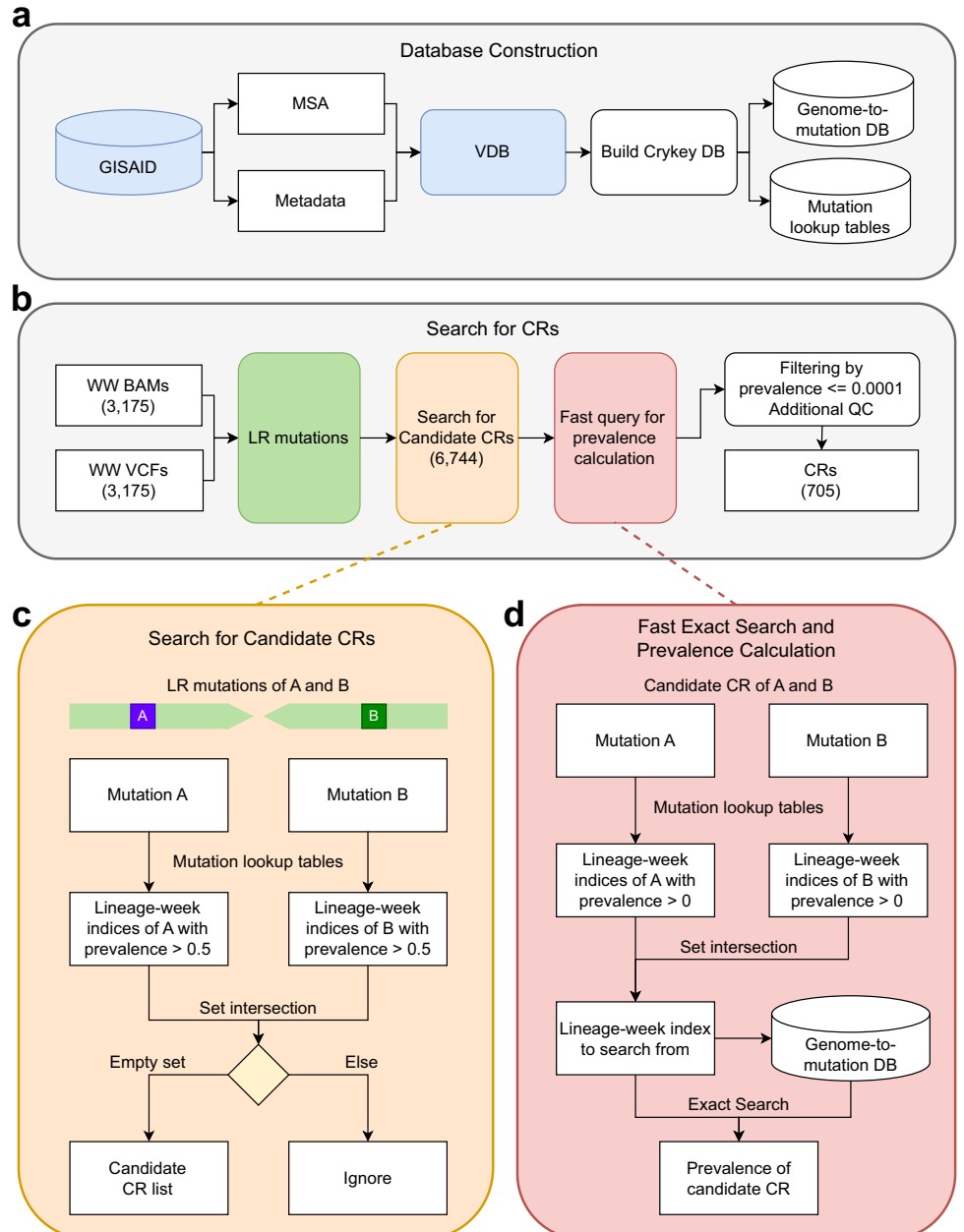

**Fig. 1 | Workflow and algorithms of Crykey. a** Crykey constructs a genome-to-mutation database and a set of mutation lookup tables using GISAID sequences and metadata. **b** Crykey searches for two or more mutations located on the same read or read-pair and uses the mutation lookup tables to identify whether the linked-read mutations represent a candidate CR. Then, each candidate CR is queried against the genome-to-mutation database to calculate its prevalence rate; if they meet the indicated thresholds they are then considered a CR. The count of wastewater samples, candidate CRs, and CRs after filtering used in this study are shown in parentheses. **c** Algorithm to search candidate CRs, with an example of a read-pair containing mutations A and B. **d** Algorithm for the fast exact search for prevalence calculation, with an example of a candidate CR containing mutations A and B.

associated with Omicron. Likewise, the consensus genomes for most of the Houston clinical samples carrying CRs (all but CR2) were identified as Omicron, mainly BA.1.1 and BA.1.15 (Fig. 7). The clinical prevalence of the Omicron CRs increased as the Omicron variant spread in the city, as reflected by both the viral load in the wastewater (Fig. 4a) and the number of sequences from Texas in GISAID (Fig. 4b). In contrast, CR2 was detected in the wastewater only during the first 2 weeks of the sampling period, while also detected with very low prevalence in the clinical samples during weeks 1–6 (Fig. 7). We also queried the number of sequences with each amino acid change associated with CR1-CR12 all over the world using outbreak.info[30,31]. As expected, the consensus-level mutations are often found in millions of SARS-CoV-2

sequences, and the mutations with low AF are found in a much lower number of sequences, ranging from a handful to thousands (Supplementary Table 2). The compendium of evidence for CR1-CR12 provides a mixed picture of factors driving the rare LR mutations comprising these CRs. Low allele frequency, high prevalence, and geographic discordance cast doubt on these CRs representing legitimate cryptic lineages.

However, CR8 exhibited a different pattern. We first detected CR8 in clinical samples in the 1st week at a low prevalence rate. As the prevalence rate in clinical samples increased, we could detect CR8 in wastewater on the 3rd week (Fig. 8), from samples sequenced using distinct protocols. CR8 consisted of two mutations, C10449A and

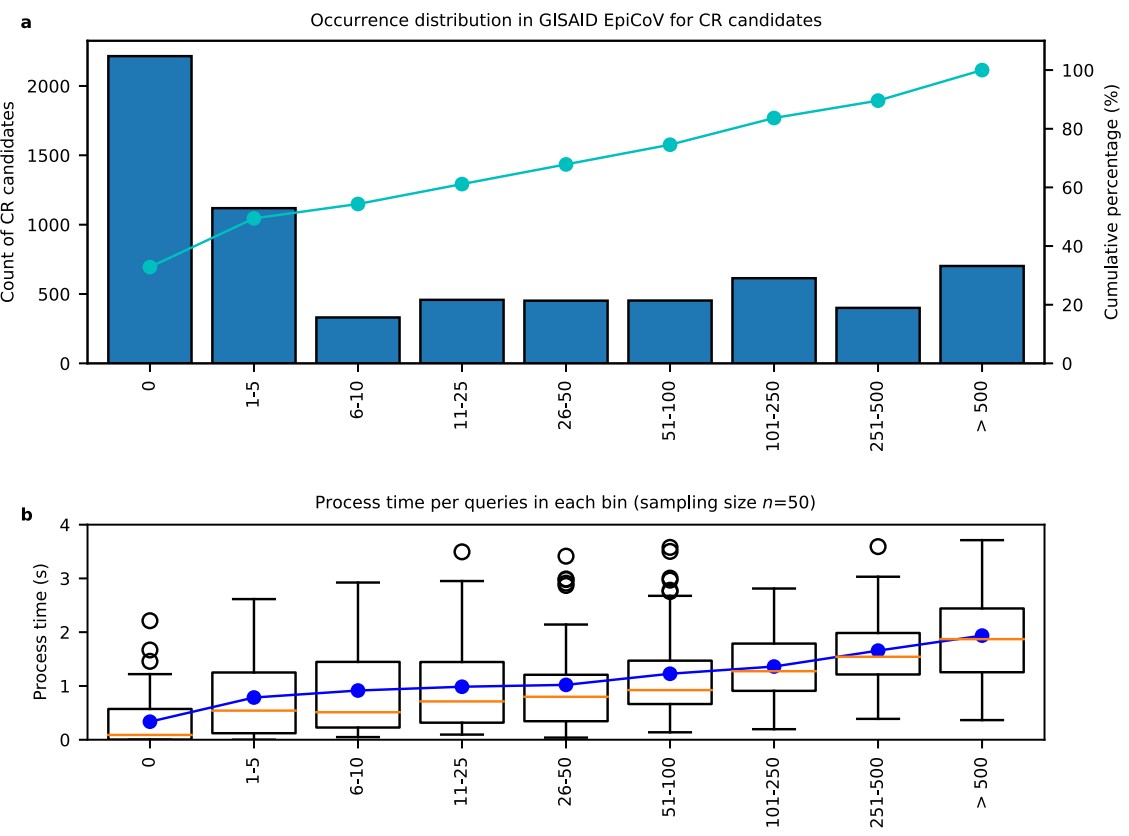

**Fig. 2 | Occurrence distribution and query time of candidate CRs found in Houston wastewater samples.** The candidate CRs identified in the samples are partitioned into bins based on their prevalence in the GISAID database. **a** *y*-axis shows the number of candidate CRs in each bin (*n* = 2215, 1119, 331, 458, 452, 453, 614, 400, 702). Cumulative percentages are plotted with a solid line on the second *y*-axis. **b** shows the process time of each bin in the box plot. For each bin, *n* = 50 independent measurements were randomly selected. The box plot includes both median lines (solid) and the box bounds the interquartile range (IQR). The Tukey-style whiskers extend from the box by at most 1.5× IQR. The outliers are shown. The average process time of each bin is shown as a solid blue line. Source data are available for this figure and are provided in the Source Data file.

T10459C. C10449A was a consensus-level mutation for Omicron strains, and it had an individual mean AF close to 1 in both wastewater and clinical samples (Fig. 8a, e). The prevalence rate of C10449A alone gradually increased in the first 3 weeks of detections starting from the week of 2021-12-06, until the prevalence rate reached 1, and the pattern was consistent in both wastewater and clinical samples; on the other hand, mutation T10459C was present as a low-frequency mutation with individual mean AF close to 0.02. The prevalence rate of T10459C alone in clinical samples increased in the first half of the sampling period, reached a peak at week 4, and then decreased in the second half of the sampling period (Fig. 8a, e). Since CR8 contained both a consensus-level mutation C10449A and a low-frequency mutation T10459C, both mean AF and prevalence rate of the co-occurring mutations followed the pattern of the T10459C (Fig. 8b, f) and as the prevalence rate of CR8 in clinical samples increased, we started to detect it in wastewater on week 3 as well. The average number of reads that span CR8 regions are shown in Fig. 8c, g as coverage and Crykey is sensitive enough to detect CR8 in wastewater, given that the coverage of wastewater samples was much lower than in clinical samples (Fig. 8d).

To assess whether geographic patterns at a national level were associated with these CRs, we processed nearly 9000 clinical samples collected outside of Texas over the same 8-week period (between 2021-12-06 and 2022-01-31; Supplementary Fig. 2). CR5 was detected across clinical samples from Maryland (very high prevalence) and Massachusetts (low prevalence) (Supplementary Fig. 6a). CR8 was detected in Maryland again at a very high prevalence (Supplementary Fig. 6b). In addition, we identified five additional CRs shared across

clinical samples from Houston and Maryland (CR3, CR4, CR7, CR9, CR11). Note that the distribution of the PANGO assignments for samples containing CR5 and CR8 differed between states. Although both CRs are associated with Omicron, Houston was dominated by BA.1.15, while Maryland and Massachusetts had a much higher proportion of BA.1.1 and BA.1.17, and Maryland had a much higher proportion of BA.1.18 and BA.1.20 as well.

## Discussion

Wastewater monitoring for SARS-CoV-2 has been widely used to complement clinical genomic surveillance during the COVID-19 pandemic[13,32]. A recent study claims to have identified cryptic SARS-CoV-2 lineages in the wastewater that went undetected in the clinic, leaving an open question about the origin of these CRs[19]. Furthermore, most cryptic lineage detection methods require ultra-deep sequencing or combining data from both long and short reads and can not be applied with commonly used wastewater surveillance protocols. Our contribution centers on a novel detection tool, Crykey, designed to identify rare linked-read mutations in wastewater using sequencing data. Specifically, Crykey leverages an optimized database lookup for the co-occurrence of mutations that are present on the same reads or read pairs and to detect the presence of CRs. Our method is fully compatible with standard mutation calling pipelines for SARS-CoV-2 and considers CRs defined by mutations that may occur across the entire SARS-CoV-2 genome. To demonstrate the efficacy of our new computational tool, we applied CryKey to >3000 wastewater samples from Houston.

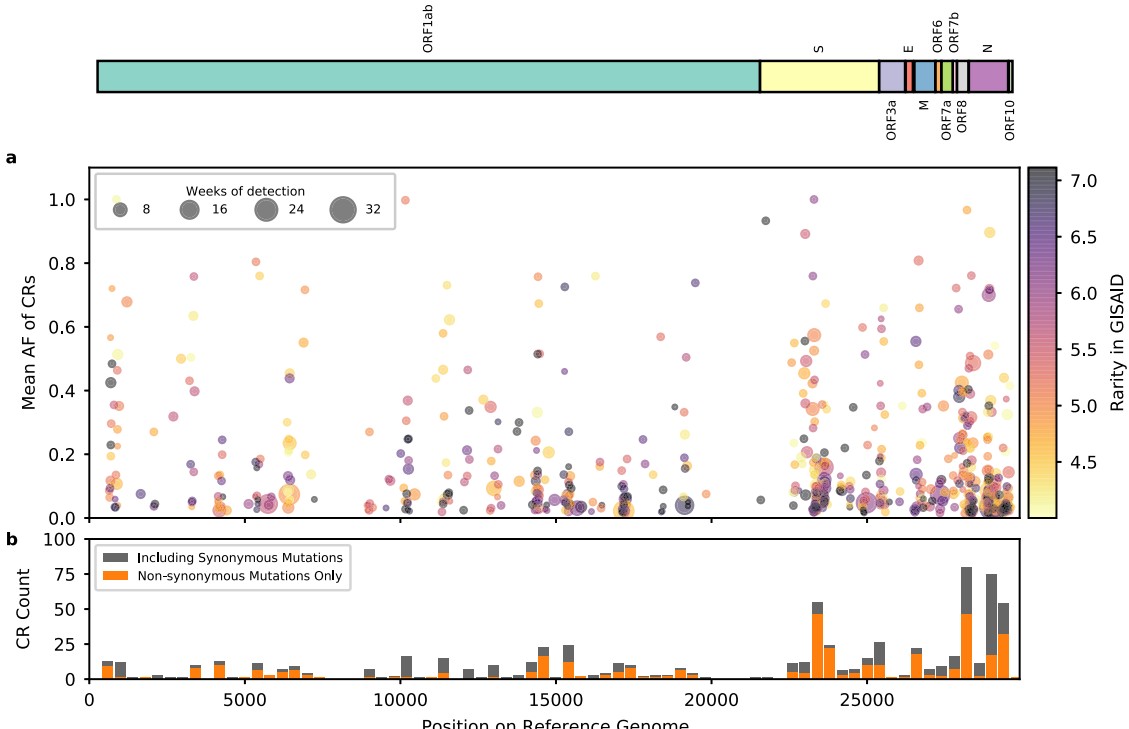

**Fig. 3 | Distribution of CRs found in Houston wastewater.** In both **a** and **b**, the locations of CRs on the SARS-CoV-2 reference genome found in Houston wastewater samples are shown on the *x*-axis, with SARS-CoV-2 ORFs shown above the figure panels. In panel **a**, each CR is represented by a colored dot, the *y*-axis indicates its mean AF in the wastewater sample, and the color indicates its rarity, defined as $-\log10((n+1)/N)$, where $n$ is the number of genomes supporting the CR in the GISAID EpiCoV database, and $N$ is the total number of sequences in GISAID; the larger the number the rarer the mutation in GISAID. The darker color suggests that the CR is rare or unreported. The size of the dot shows the number of weeks the CR was detected. Larger dots indicate the CR persisted longer in the community. Panel **b** is a histogram showing the count of CRs found in different 400 bp regions of the reference genome. CRs containing exclusively non-synonymous mutations are marked in orange, and the CRs containing at least one synonymous mutation are marked in gray. Higher bars indicate that more CRs were found in the associated region. Source data are available for this figure and are provided in the Source Data file.

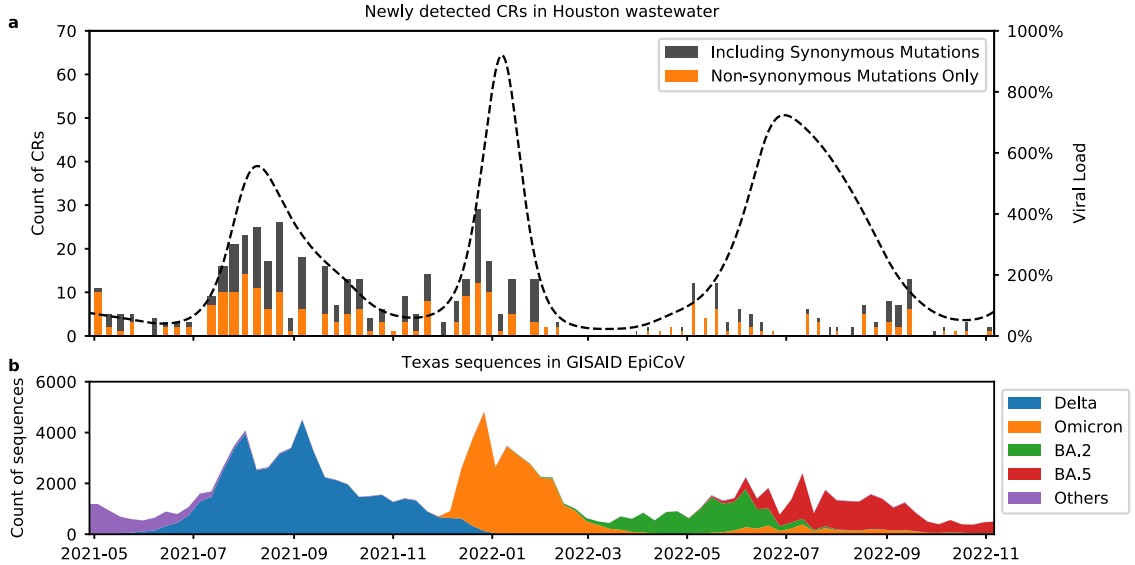

**Fig. 4 | CRs and viral load in Houston wastewater.** In both panels **a** and **b**, the *x*-axis shows the dates from May 2021 to November 2022. Panel **a** shows the number of CRs (left *y*-axis) newly detected in Houston wastewater per week as bars. The proportion of CRs containing only non-synonymous mutations is indicated in orange, while the remainder is in gray. The width of the bar indicates the average breadth of genome coverage across all WWTP, ranging from 0.02 to 0.74. The normalized viral load in wastewater (right *y*-axis) (based on the viral load from samples collected on July 6, 2020 in Houston) is shown as a dotted line. Panel **b** shows the number of SARS-CoV-2 sequences in the GISAID EpiCoV database from Texas, USA per sampling week. Color corresponds to their PANGO lineage assignments. Omicron lineages other than BA.2, BA.5, and their descendants are combined and denoted as "Omicron". Source data are available for this figure and are provided in the Source Data file.

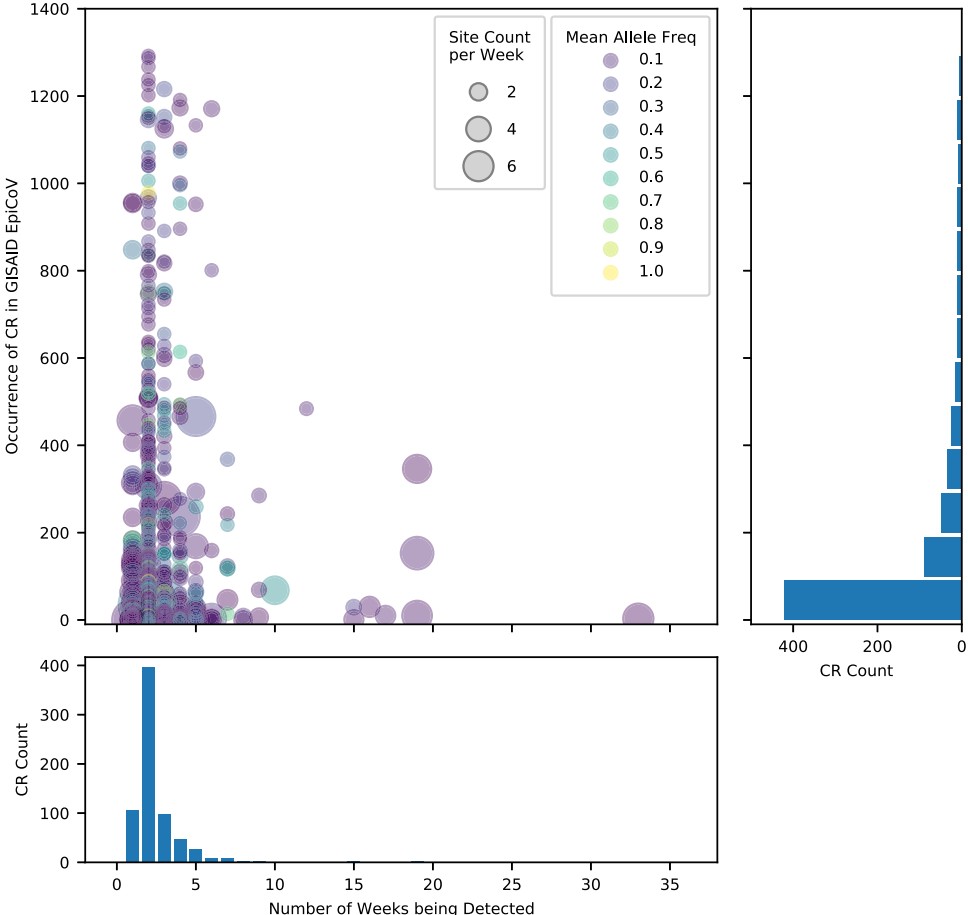

**Fig. 5 | Persistence and Occurrence of CRs found in Houston wastewater.** Each CR is represented by a dot, with the size of the dot indicating the mean count of the wastewater treatment plants the CR was detected at each week, and the color of the dot indicating mean allele frequency. The histogram on the bottom shows the number of weeks that the unique CRs have been detected and their associated counts. The histogram on the right shows the rarity of unique CRs in terms of occurrence in GISAID and their associated counts. Source data are available for this figure and are provided in the Source Data file.

By examining 3 years of wastewater sequencing data and 8 weeks of local clinical surveillance data, our goal was to demonstrate the potential of Crykey to provide a finer-grain view of the emergence of potential cryptic lineages within Houston. Our results suggest that the number of detected CRs in wastewater relates to the shift of dominant VOCs in the region. We also showed that twelve CRs were found in wastewater and clinical samples from the same time period. Future work is required to validate CRs as they emerge and to discern between potential systematic biases and legitimate CRs. In particular, cases where the CR is highly prevalent in clinical samples but at a low frequency within each individual and cases when all the mutations comprising a CR exhibit similar allele frequencies (which could represent read mapping or alignment error caused by indels, or strong evidence for a novel cryptic lineage given the very low likelihood of multiple errors being introduced on a single read from a high-fidelity sequencing platform), and combinations of consensus level established mutations observed with single low-frequency mutations (which could represent the emergence of a novel SARS-CoV-2 lineage or also represent an error co-located with a characteristic mutation from a PANGO lineage) warrant further investigation.

### Interrogating outlier CRs with Crykey

The emergence of the CRs coincided with the spread of new variants of concern. We observed an increased number of CRs being detected in Houston wastewater during the emergence of Delta (July 2021) and Omicron (December 2021). While 85% of the CRs lasted for less than 4 weeks, we also observed some CRs that persisted for more than

10 weeks (Fig. 5). Notably, CR12 was detected across multiple WWTPs in Houston for 33 weeks (Fig. 6). CR12 contains two LR mutations, A29039T and G29049A, which cause K256* and R259Q amino acid changes on the N gene. The combination of these mutations is rare; only three entries in GISAID contain both mutations, and none of these originated from the United States. Previous work has shown that N:K256 is one of the eight lysine residues in the protein N of SARS-CoV-2 that is likely to be directly involved in RNA binding[33]. A29039T generates a stop codon that may affect the linker region, suppressing the immunogenic domain of the nucleocapsid protein, which might help the vaccine escape[33,34]. N:R259 belongs to one of the identified guanosine triphosphate binding pockets, and is well-conserved in multiple human coronaviruses, including NL63, 229E, HKU1, OC43, as well as MERS, and SARS-CoV-1[35], the N:R259Q mutation has been reported multiple times at low prevalence rates in several SARS-CoV-2 lineages, likely representing a hotspot mutation mostly belonging to the Delta variant[31]. A previous study suggested that the nucleocapsid protein of SARS-CoV-2 is flexible and dynamic, and CR12 happens to be located on one of the predicted disordered regions of the N gene[36].

### Potential origins of cryptic mutations: cryptic lineages, poorly understood biological signal, or systematic noise?

The precise origin of the cryptic mutations we found in Houston wastewater remains an open question. One could think of five possible scenarios: (1) they represent rare circulating SARS-CoV-2 lineages that went un-sampled or under-sampled in the clinical samples, (2) they exist as intra-host mutations from the population that have high

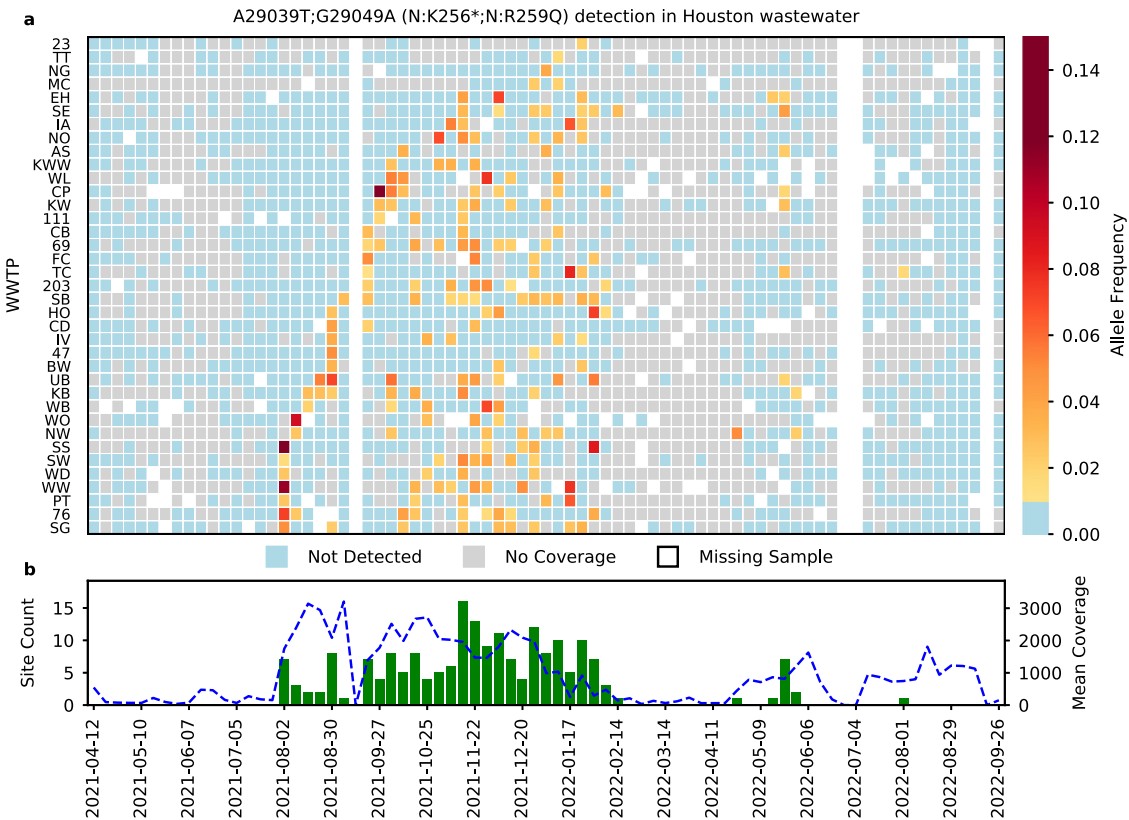

**Fig. 6 | Persistence of CR12 (A29039T-G29049A) in Houston wastewater.** For both panels, the *x*-axis shows time. In panel **a**, rows correspond to wastewater treatment plants (WWTPs) sampled, with the cell color indicating the mean allele frequency (MAF) of the mutation set (cells with MAFs below 0.01 are colored as blue and labeled as "Not Detected". Samples with coverage below 10x are marked in gray and denoted as "No Coverage". Missing samples are marked in white. In panel **b** the bars indicate the number of WWTPs in which CR12 was detected, per week (left *y*-axis). The dotted blue line indicates the mean coverage of the wastewater samples with coverage above 10x, per week (right *y*-axis). Source data are available for this figure and are provided in the Source Data file.

enough prevalence to be detected in wastewater, (3) they represent signal from SARS-CoV-2 transcription such as subgenomic mRNAs, (4) they are spillover from an unidentified animal reservoir, or (5) they are technical artifacts from environmental degradation, sample preparation, or sequencing.

A possible explanation behind CRs in the wastewater not being captured by clinical surveillance is low community prevalence rates[19,20,24]. As only a small portion of the SARS-CoV-2 infections are sequenced, transient cryptic lineages are likely to be missed by clinical surveillance. Clinical data also suffers from sampling bias, where people with severe symptoms and access to healthcare resources are more likely to be represented in the databases. Figure 5 shows that most of the cryptic mutations detected in Houston wastewater were only found over 1 to 3 weeks, and these short-duration cryptic mutations may represent those not captured by clinical testing. In support of this hypothesis, we found a subset of the cryptic mutations supported by reads from clinical sequencing.

However, we observed several cryptic mutations with a high prevalence and low intra-host AF in the clinical samples. Indeed, it is common to only report consensus-level mutations (i.e., mutations with allele frequencies greater or equal to 0.5), or consensus genomes/ assemblies to the public databases such as GISAID. As a result, although CRs might be sampled, they will remain unreported. A recent study has shown that molnupiravir treatment can induce de novo mutations in multiple individuals[37]. Still, it is not clear whether the cryptic mutations found in clinical samples are tied to therapy-related lineages. We also observed cases where a CR persisted for multiple weeks in wastewater samples but had little to no trace in clinical samples. Why these CRs were not captured by clinical surveillance

remains unknown. As a possible explanation for this second scenario, previous studies have suggested that cryptic lineages may be carried by non-human hosts[19], especially for those that persist for very long periods[38]. Given that we lack representative genomes from non-human hosts during the time frame of our results, we cannot investigate the plausibility of this scenario.

Temporally linked CRs (especially those appearing and disappearing within a few weeks) provide reliable evidence for legitimate novel cryptic lineages. Additionally, CRs that contain multiple low-frequency mutations in a single read (and all supported by 5 or more reads) contrast themselves with CRs that contain a mutation shared with a circulating PANGO lineage and a companion low-frequency mutation (or mutations). On the surface, low mean allele frequencies combined with high prevalence rates in clinical samples raise some concern regarding their validity, especially given the lack of plausible explanation for the transmission/community spread of low-frequency intra-host mutations (Supplementary Fig. 5).

To investigate the potential origin of CRs, we evaluated a dataset of 5060 clinical samples collected within Greater Houston from 2021-12-06 to 2022-01-31. Our results suggest that CRs detected in wastewater could be related to intra-host low-frequency co-occurring mutations in clinical samples (Figs. 7 and 8). The unusually long life span of CR12 and its high prevalence at low intra-host AF in clinical samples suggest that it could be a previously unreported artifact. However, examining over 7000 clinical samples sequenced with the PacBio HiFi system, and testing multiple read aligners, we can likely rule out that CR12 is primarily related to primer artifacts, sequencing technology-dependent artifacts, and sequence alignment errors.

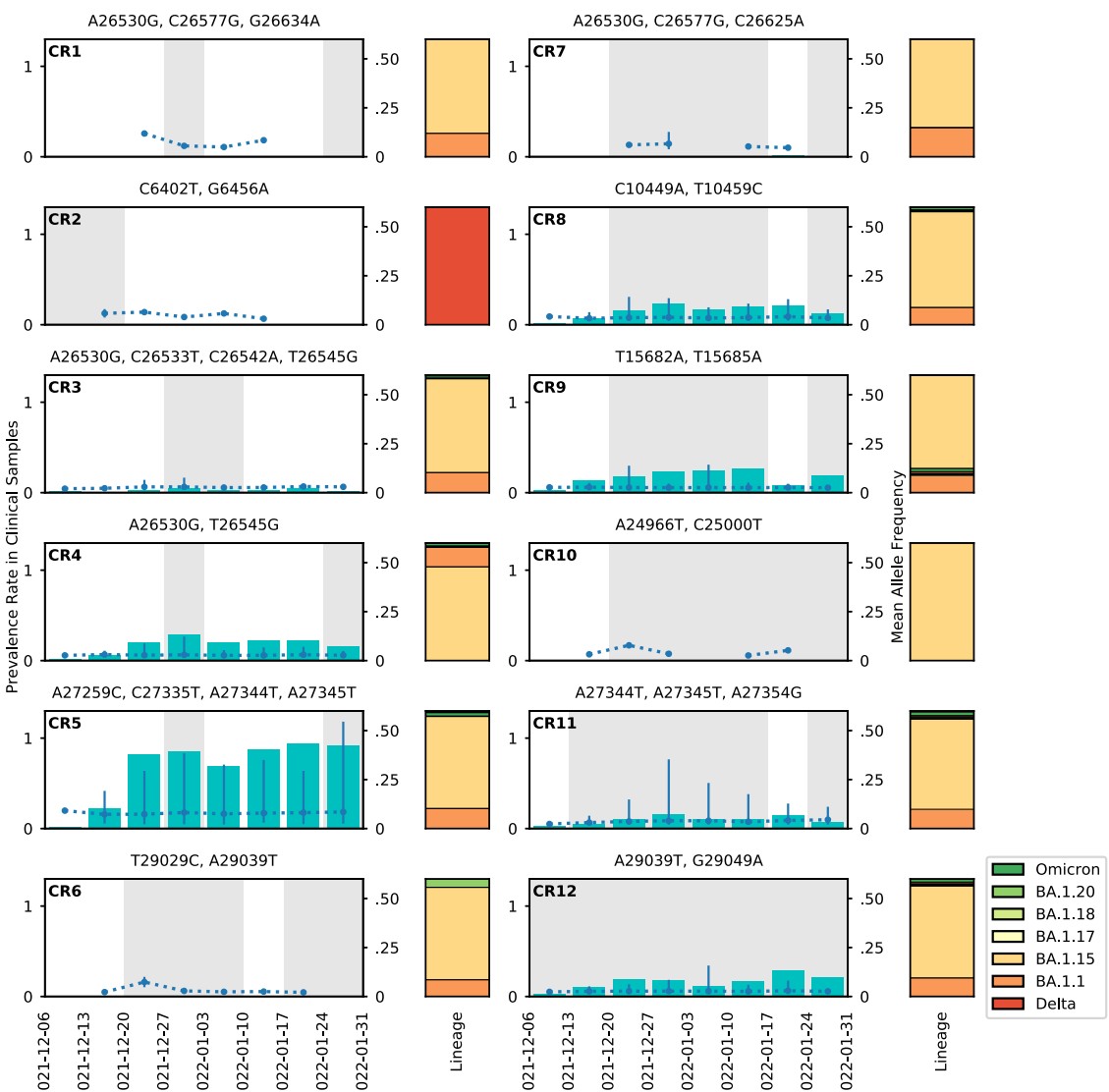

**Fig. 7 | CRs detected in clinical samples from Greater Houston.** The mutation combinations for each wastewater CR are shown at the top of each panel. Cyan bars indicate the prevalence (left *y*-axis). The dotted blue lines (right *y*-axis) indicate the mean AF (right *y*-axis) of the CR in the clinical samples. The number of biologically independent clinical samples that the CR is prevalent are *n* = 5; 9; 163; 1115; 3830; 14; 20; 929; 1079; 7; 581; 845 for CR1 to CR12. The exact *n* for each CR per week is reported in the Source Data file. The error bars are defined as the range of the observed AF (right *y*-axis), with the top of the error bars showing maximum observed AF, and the bottom of the error bars showing minimum observed AF. The areas shaded in gray indicate the periods during which the CR was detected in the wastewater. The stacked bars to the right of the panels show the distribution of the PANGO lineages of the consensus genomes of clinical samples with CRs with each category shown in different colors. All Delta genomes are combined. All Omicron genomes other than BA.1.1, BA.1.15, BA.1.17, BA.1.18, and BA.1.20 are combined and denoted as Omicron. Source data are available for this figure and are provided in the Source Data file.

Finally, cryptic mutations could represent some type of systematic bias. Even though we have taken extreme care to filter out known sources of artifacts and have observed them across different sample types, amplicon panels, read mappers, and sequencers, we cannot rule out unknown systematic artifacts or biases leading to CR detection. Given there is no single mutational pattern observed (they can be comprised by multiple low allele frequency mutations that are short duration or very long duration, as well as contain mutations that pair an established consensus-level mutation from a VOC with a transient low-frequency mutation), explanations for each of these patterns and their variability over time requires further investigation. Indeed, the goal in developing Crykey was to provide an efficient and sensitive tool for interrogating cryptic mutation patterns over time and geography, hoping to shed light on their origin and facilitate the identification of artifacts.

**Open challenges in tracking potential cryptic lineage mutations in wastewater**

One of the key challenges in reliably detecting CRs in wastewater is the quality of the samples[39,40]. As shown in Fig. 4, the number of newly emerging CRs follows the same pattern as the viral load until June 2022, where the samples collected afterward had worse quality regarding breadth of coverage. The performance of Crykey is limited because the samples did not have enough sequencing depth across most of the regions of the SARS-CoV-2 genome during those weeks. Due to the inherent limitations of short-read sequencing platforms that generate 100–200 bp reads, protocols used for sequencing, and the fragmented state of the viral RNA in wastewater, there is a natural limit on the genomic span of the cryptic mutations we can use. Indeed, the degradation of genetic material in wastewater impacts the sequencing quality of the sample while introducing noise for rare

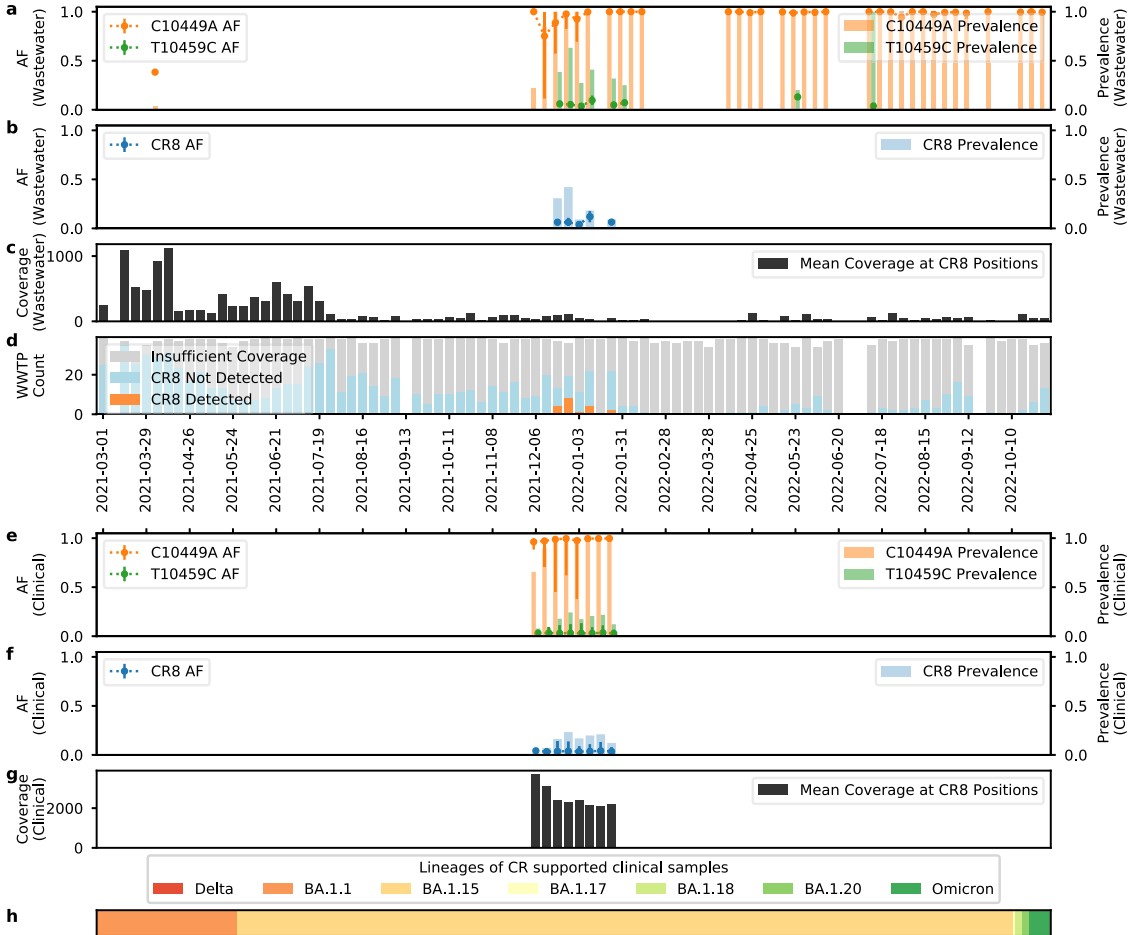

**Fig. 8 | CR8 detected in wastewater and clinical samples from Greater Houston.**
Figure **a**–**d** are information on CR in wastewater each week, with sample collection date shown in *x*-axis. **a** shows the prevalence rate as bars (left *y*-axis, same for **b**, **e**, **f**) of each individual mutation within CR in wastewater samples, and the mean AF (right *y*-axis, same for **b**, **e**, **f**) of each individual mutation presented as a dotted line of $n = 222$ biologically independent samples for C10449A (shown in orange) and $n = 39$ for T10459C (shown in green). The error bars (right *y*-axis, same for **b**, **e**, **f**) are defined as the range of AF, with the top of the error bars showing maximum observed AF, and the bottom of the error bars showing minimum observed AF. **b** shows the prevalence rate of the CR in wastewater samples, and the mean AF of CR ($n = 19$). **c** shows the mean coverage at CR8 locations. **d** shows the sample qualities and Crykey detections, with samples of insufficient coverage colored in gray, samples of CR absent colored in blue, and samples of CR detected colored in orange. Figure **e**–**h** are information on CR in clinical samples of Houston for 8 weeks

of the sampling period. For figure **e**–**f**, the sample collection date is shown in *x*-axis. **e** shows the prevalence rate as bars of each individual mutation within CR in clinical samples, and the mean AF of each individual mutation is presented as a dotted line with $n = 4937$ for C10449A (shown in orange) and $n = 981$ for T10459C (shown in green). **f** shows the prevalence rate of CR in clinical samples, and the mean AF of CR ($n = 929$). **g** shows the mean coverage at CR8 locations. **h** shows the distribution of the PANGO lineages of the consensus genomes of the clinical samples with CR. Delta genomes are not found in any of the samples. All Omicron genomes other than BA.1.1, BA.1.15, BA.1.17, BA.1.18, and BA.1.20 are combined and denoted as Omicron. For figure **a**–**c**, and **e**–**h**, wastewater and clinical samples with insufficient coverage (<10x at CR location) are excluded from the analysis. Source data are available for this figure and are provided in the Source Data file.

mutation detection[41,42]. Furthermore, genomic regions corresponding to sequencing primers or adapters create coverage gaps (regions without read support) along the genome and pose a challenge for identifying CRs that span longer regions. However, these limitations could be addressed using long-read sequencing if sample manipulation and extraction procedures allow intact longer RNA fragments to be recoverable from wastewater samples.

Crykey represents an efficient and easy-to-use tool designed to rapidly and comprehensively find cryptic mutations across thousands of wastewater samples. We applied Crykey to detect numerous CRs in Houston, some persisting for months. The concept of searching for rare LR mutations inside of a viral genome that has never or rarely been reported is generalizable, and Crykey is not limited to SARS-CoV-2. Crykey can be expanded and applied to multiple pathogens, such as influenza viruses, as long as the pathogen has an established database of genomic sequences[43,44]. We hope our findings will help promote

community-wide discussion on best practices for cryptic lineage tracking in wastewater.

## Methods

Crykey is a computational method for identifying cryptic mutations representing potential cryptic lineages (CRs) in wastewater samples on a full-genome scale. We identify cryptic mutations as sets of two or more co-occurring mutations in the same sequencing read that appear in 5 or more reads, with a minimum AF of 0.01 in two or more samples and observed together in less than 0.01% of the GISAID EpiCoV genomes (up until 10/21/2022). The workflow of the Crykey pipeline can be divided into 3 steps, including database construction, sample processing to find CR candidates, and rarity calculations for each candidate found in the previous step. Crykey first builds mutation look-up tables and a genome-to-mutation database using the full GISAID's EpiCoV database (Fig. 1a) and then searches for CRs (Fig. 1b).

Specifically, Crykey first extracts LR mutations from the alignment and searches for CR candidates by querying the mutation look-up table (Fig. 1c). Then, each CR candidate is queried against a pre-built database to check if it is novel or rare in terms of prevalence to create candidate CRs (Fig. 1d). Candidate CRs are then passed through rigorous filters to nominate a subset as detected CRs. Due to the optimized database structure that partitions the mutation prevalence information according to the associated PANGO lineage/lineages for a given time period, Crykey is highly efficient and can easily scale to thousands of samples. We will now provide specific details regarding the filtering steps and analysis methods used in this manuscript.

## Candidate CR lineage determination

The database used in Crykey is built based on the multiple sequence alignment (MSA) generated by the GISAID EpiCoV database. We extracted the mutations for each SARS-CoV-2 genome in the MSA using vdb with the command *vdbCreate -N input.msa*[45]. We then trimmed the list of mutations associated with each genome sequence with the vdb *trim* command. Combining the lineage assignment of each genome sequence in the metadata, we calculate the prevalence rate of each mutation in each of the known lineages of SARS-CoV-2, as well as build a mutation database containing mutation information for each individual SARS-CoV-2 genome. The results shown in this manuscript are based on the 13,875,227 individual genome sequences and their associated metadata available on GISAID up to 2023-01-05, via gisaid.org/EPI_SET_240326gr (Supplementary Table 3).

To identify candidate cryptic lineages, Crykey first builds a default mutation lookup table where each mutation in GISAID is associated with a set of lineages and specific weeks (based on sample collection date) of occurrence in GISAID, regardless of prevalence rate. A second mutation lookup table is built at the same time where only mutations with a prevalence rate greater than 0.5 are stored, which allows us to perform a fast query on whether a set of SNPs belongs to any of the SARS-CoV-2 genomes in a given time period.

Then, for a given sample, Crykey takes its associated alignment file (BAM) and mutation calling output (VCF) file as input. It first extracts and filters the mutations from the VCF file with a user-defined minimum depth of coverage (default: 10) and a user-defined minimum allele frequency (default: 0.02). Then, it annotates each mutation with snpEff and removes mutations not in the coding region. For each sample, Crykey searches through the BAM file and extracts read pairs that contain a combination of two or more mutations. Using the mutation lookup table of prevalence rate greater than 0.5, Crykey can quickly identify whether the mutation combination from the read may belong to a single lineage of a certain week by using set intersection. By the pigeonhole principle, if the intersection is non-empty, it is guaranteed that the mutation combination has been reported to the public database. If the intersection is an empty set, we consider the combination as a candidate CR.

## Classifying candidate CRs from Houston wastewater

For each candidate CR, we evaluate the occurrence, temporal patterns, and the set of mutations comprising a CR. First, an exact search is performed by querying the mutations in the lineage against the default mutation lookup table and the mutation database. Using the default mutation lookup table, the lineage and specific week of the co-occurrence of all mutations in the set can be quickly determined and allows Crykey to minimize the search space while querying the mutation database, searching for exact assemblies containing candidate CR. Crykey outputs a complementary report on whether or not the CR candidates found in the sample are truly novel, which means no sequences in the database support the combination, or if the CR candidates are rarely seen in the database. If the CR candidate can be found in the database, Crykey reports the number of sequences found containing the CR candidate in each lineage and the total number of sequences in those lineages.

In our experiment, we applied both within-sample and cross-sample filtering on the candidate CRs before nominating them as CRs. To do this, we filtered each candidate cryptic lineage by only keeping the sets with supporting reads for each mutation in a CR above a minimum threshold (default: 5), and allele frequency above a minimum threshold (default: 0.01). The cross-sample filtering was based on the minimum number of WW samples supporting the candidate CR (default: 2). CR candidates with a prevalence rate above 0.0001 in the GISAID database were also removed. In the end, we excluded the CR candidates that contain mutations located between reference genome positions 1 to 55 and 29,804 to 29,903, and masked CR candidates that contain mutations located on 25 sites between 56 and 29,804 based on suggestions from previous studies, as those locations are highly homoplasic and mutations are likely to be recurrent artifacts[46,47]. Candidate CRs that pass the above filters are nominated as CRs and used as the foundation for potential cryptic lineage search in clinical samples. The use of multiple filters on the candidate CRs promotes the CR identification process to be conservative. The selection of thresholds and filters was informed by prior studies[46,48–53].

## Cross-referencing CRs in clinical samples

The samples from Houston were downloaded from the NCBI SRA database under the BioProject PRJNA764181[54]. The sequencing details of the dataset can be found in our previous study[8]. We collected all 5060 SRA samples with collection dates between 2021-12-06 and 2022-01-31 without additional filtering. In addition, 8,969 out-of-state SRA samples were selected from the NCBI database (PRJNA686984). We first performed a standard read mapping process, including quality control. The sequencing reads from the clinical samples were first filtered using fastp v0.23.2 with parameters "--cut_front --cut_tail --cut_window_size 4 --cut_mean_quality 25 --qualified_quality_phred 25 --unqualified_percent_limit 40 --n_base_limit 5 --length_required 15 --low_complexity_filter --complexity_threshold 30" to remove low-quality bases and low-quality reads[55]. Then the filtered reads were aligned to the reference genome of SARS-CoV-2 (NCBI Reference Sequence: NC_045512.2) with bwa mem v0.7.17-r1188 with default parameters[29,55]. The alignment files were sorted and indexed with samtools v1.14[56]. Mutation calling was done using lofreq v2.1.5 with command "lofreq call --no-default-filter --call-indels", and then filtered with the command "lofreq filter --cov-min 20 --af-min 0.02 -b fdr -c 0.001"[57]. Consensus genomes were generated, and PANGO calling was done using pangolin v4.2 with default parameters[58].

After read mapping, the BAM files and the VCF files are collected for searching for CRs detected in wastewater samples. 20 wastewater CRs detected during the 8-week sampling period were selected for testing. 10 of the 20 wastewater CRs occurred in 2 of the 8 weeks, representing CRs with a short burst pattern; when cross-referencing with the clinical sample data, we selected short-lived CRs detected in most wastewater treatment plants. The rest of the 10 wastewater CRs we selected for the query had the longest clinical sampling period, ranging from 4 to 8 weeks of occurrence, representing CRs with a long-lasting detection pattern in both wastewater and clinical samples.

By using the alignments in the clinical samples, we counted the total number of reads spanning the regions that the CRs contained and counted the number of reads supporting all mutations from the CRs at the same time. 5 bases towards both ends of the reads were ignored to avoid noise caused by sequencing errors. The allele frequency of a potential cryptic lineage was calculated as the number of CR supporting reads over the number of total reads covering those positions.

During the analysis, we further filtered the results, and samples with CRs with less than 5 supported reads or with AF less than 0.02, or any of the mutations within the CR missing from the variant calling are considered as CR absent. We counted forward and reverse read fragments that do and do not fully support all cryptic mutations, and calculated both the *p*-value of the two-sided Fisher's exact test and

strand bias scores described in the previous studies[59,60]. Samples with reads containing strain bias score greater or equal to 1 and a *p*-value of Fisher's exact test less than 0.05 are also considered as CR absent.

All wastewater samples and clinical samples with insufficient coverage for a CR (the number of reads that cover all mutation positions of a CR is less than 10) are excluded from the calculation of AF, prevalence rate, and coverage in Fig. 7 and Supplementary Figs. 3–5. The AFs of individual mutations from the CR are extracted from the variant calling results. The AF of CR is calculated as the number of reads that contain all mutations of a CR over the number of reads that cover all mutation positions of a CR. The prevalence rate is calculated as the count of CR/mutation detected samples over the count of samples with sufficient coverage.

### Validating CR12 (A29039T-G29049A) CR with PacBio clinical samples
The samples were downloaded from the NCBI SRA database under the BioProject PRJNA716984. We subsampled 7,113 SRA runs with sample collection dates between 2021-11-06 and 2022-03-21, including 2,458 samples collected in Texas and 4,655 samples from 50 other regions (49 US states and Puerto Rico). Samples with missing metadata (location or sample collection date) were excluded. The reads were aligned to the reference genome of SARS-CoV-2 (NCBI Reference Sequence: NC_045512.2) with minimap2 using map-pb preset[61]. The alignment files were sorted and indexed with samtools v1.14. The number of supporting reads and the depth of coverage are calculated using the same method described in the previous section.

### Reporting summary
Further information on research design is available in the Nature Portfolio Reporting Summary linked to this article.

## Data availability
Source data is provided with this paper and has been deposited with DOI: 10.5281/zenodo.10934124. All sequencing data supporting the findings of this study is publicly available. Houston wastewater datasets are available for download via NCBI BioProject PRJNA796340. Houston short-read clinical datasets are available for download via NCBI BioProject PRJNA764181. Non-Texas short-read clinical datasets are available for download via NCBI BioProject PRJNA686984. The PacBio SARS-CoV-2 clinical datasets are available for download via NCBI BioProject PRJNA718231. Source data are provided in this paper.

## Code availability
The source code for Crykey is publicly available at https://github.com/treangenlab/crykey, and we used version 1.0.0 of Crykey for the result and analysis presented in this manuscript[62]. The code used for analysis and figure generation used in this study can be found at https://github.com/treangenlab/crykey_analysis_scripts[63].

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

## Acknowledgements

We gratefully acknowledge all data contributors, i.e., the Authors and their Originating laboratories responsible for obtaining the specimens, and their Submitting laboratories for generating the genetic sequence and metadata and sharing via the GISAID Initiative, on which this research is based. The authors thank all of the GISAID contributors who provided the SARS-CoV-2 genomic data. We also thank Dr. Loren Hopkins, Dr. Kathy Ensor, Kaavya Domakonda, Rebecca Schneider, and Anthony Mulenga for their leadership and contributions to the Houston Wastewater Epidemiology system. We also thank Dr. Adolfo Lara, Roger Sealy, Pamela Brown, Ryker Penn, and Yanlai Lai (Houston Health Department), as well as Dr. Esther Lou, Lauren Bauhs, Robert Campos, Russell Carlson-Stadler, Madeline Wolken, Kyle Palmer, Whitney Rich (Rice University). This work was supported in part by the Houston Health Department. Y.L., N.S., and T.J.T. were supported in part by the C3.ai Digital Transformation Institute (C3.ai DTI), Centers for Disease Control (CDC) contract 75D30121C11180, and National Institute of Allergy and Infectious Diseases (NIAID) P01AI152999-01 award. T.J.T. was also supported by National Science Foundation (NSF) grants EF-2126387, IIS-2239114, and CNS-1338099. N.S. was also supported by the Ken Kennedy Institute Andrew Ladd Memorial Excellence in Computer Science Fellowship. L.B.S. was supported in part by the National Science Foundation (CBET 2029025), seed funds from Rice University, the City of Houston, and CDC contract 75D30122C14709. D.P. also receives support from the Spanish Ministry of Science and Innovation-MICINN (PID2019-106247GB-I00) and from Xunta de Galicia. PGG was supported by grant ED481A-2021/345 from the Consellería de Cultura, Educación e Universidade Xunta de Galicia.

## Author contributions

All authors conceived the experiments, analyzed the results, and drafted and revised the manuscript. Y.L. conducted the experiments. Y.L. and N.S. wrote the code. Y.L., P.G., L.T., and D.P. analyzed the result. Y.L., N.S., P.G., L.T., D.P., L.S., and T.T. drafted and revised the manuscript. L.S. and T.T. managed the study and supervised the project.

## Competing interests

The authors declare no competing interests.
