## [Peer Review File · Nature Communications]

Crykey: rapid identification of SARS-CoV-2 cryptic mutations in wastewaterREVIEWER COMMENTS

Reviewer #1 (Remarks to the Author):

Liu et al. described a computational tool for identifying very low frequency substitutions in the SARS-CoV-2 genome from wastewater and clinical specimens. It is based on a look-up table of mutations and searching for rare mutations in the EpiCoV database in GISAID. Results showed that hundreds of cryptic mutations were identified and these mutations persisted for several weeks in wastewater. While the use of wastewater in SARS-CoV-2 surveillance is important in early detection and identification of rare lineages, however it is unknown if these rare mutations are from natural infections or technical biases. The manuscript can be improved by addressing the following issues.

1. The rationale for the definition of cryptic mutations is unclear. What is the basis of using 5 or more reads and frequency of less than 0.01% to be considered as CRs? What is the false discovery rate when using this definition?
2. Perform validation of cryptic mutations using long-read sequencing, e.g. Nanopore, especially for CRs with low coverage depth.
3. What is the origin and timing of these CRs?
4. Does the persistent detection of CRs in wastewater reflect the community surveillance data or is it due to immunocompromised individuals shedding these viruses with CRs?
5. Compare and contrast Crykey with previous methods in terms of speed, accuracy, sensitivity, etc.
6. In Introduction, after identifying what is unknown, explicitly state the objectives of the study.
7. In Methods, why is the study period of clinical data short (8 weeks) vs. 3 years for wastewater?
8. In Results, show a figure that parallels the workflow, which tracks the initial number of sequences down to identifying RCs.
9. In Discussion, restructure the discussion such that it will address the objectives of the study in the early paragraphs.

Reviewer #2 (Remarks to the Author):

This is a well-written and clear paper that presents a novel tool that would allow the identification of and determination of the frequency of cryptic lineages among sequencing samples. The paper is significant in that it presents a solution to known problems associated with these types of variants and pays attention to the current limitations and challenges associated with the work. The tool has broad applicability to viruses and potentially to other small genomes owing to the design of the tools in that it takes into account bins of time to sample from. A possible addition would be to restrict the sampling to geographic locations owing to the geographic distributions of emerging viral variants?

I am not aware of another program that looks at the whole genome and reports specifically on these types of mutation frequencies given the definition of cryptic mutations that they describe.

Specific edits:

Line 141 need a space between Figure and 7

Line 144 are found in "A" much lower number..

Lines 149-164 the tenses throughout this paragraph are confusing - I'd prefer all current or past tense...

Line 158 reaching "a" peak

Line 164 than "in" clinical samples

Line 174 Maryland had "a" much higher proportion

Line 178 fix the reference ,

Line 259 and "testing" multiple read aligners

Line 295 extra space between cryptic and lineage

Line 298 occurring "in" less than

Line 299 extra space between) and in wastewater

Line 297-300 consider a rewrite of this whole sentence, it is confusing as written

Line 305 Then, "each"

I don't believe that additional work is needed to demonstrate the conclusions presented and the claims herein. The authors have endeavored to present all possible interpretations and have honestly considered the limitations. I do not detect any flaws in the data analysis, or the interpretation or conclusions. The methodology appears sound. I believe it meets standards.

The detail and instructions provided are likely sufficient to support testing and implementation of the method. As with many other algorithms and tools, the use of the tool with likely reveal some needed modifications in future derivations.

LEGEND

Author responses to editor/reviewers in **bold black**
Changes to the manuscript in **blue**
Location of inserted text in (parenthesis)
Added/updated display items indicated in **red**
Reviewers' comments in black

We thank the editor for providing the opportunity to submit a revised version of our manuscript. We have carefully addressed the reviewers' concerns, which have strengthened our study and better highlighted the performance of our method Crykey. Point-by-point responses to reviewer feedback are provided below.

Reviewer #1 (Remarks to the Author):

Liu et al. described a computational tool for identifying very low frequency substitutions in the SARS-CoV-2 genome from wastewater and clinical specimens. It is based on a look-up table of mutations and searching for rare mutations in the EpiCoV database in GISAID. Results showed that hundreds of cryptic mutations were identified and these mutations persisted for several weeks in wastewater. While the use of wastewater in SARS-CoV-2 surveillance is important in early detection and identification of rare lineages, however it is unknown if these rare mutations are from natural infections or technical biases. The manuscript can be improved by addressing the following issues.

We appreciate the reviewer's feedback. However, we would like to point out that Crykey identifies rare events that include multiple mutations (not necessarily low-frequency) co-occurring on the same SARS-CoV-2 genome. Therefore, the main focus of the manuscript is on cryptic lineage (CR) detection instead of low-frequency variant calling.

R1Q1. The rationale for the definition of cryptic mutations is unclear. What is the basis of using 5 or more reads and frequency of less than 0.01% to be considered as CRs? What is the false discovery rate when using this definition?

For a set of single nucleotide variants (SNVs) to qualify as a cryptic lineage (CR), the following conditions have to be met:

- 1. Each SNV must be called independently by the variant caller (lofreq) with a minimum allele frequency (AF) of 2%.**
- 2. All SNVs must co-occur on a sequencing read, and there must be at least 5 such reads supporting every SNV.**
- 3. The AF of the co-occurring SNV set, calculated as the number of supporting reads divided by the number of reads spanning the region, must be 1% or higher.**
- 4. The set of co-occurring SNVs is observed in at least 2 wastewater samples.**
- 5. The set of co-occurring SNVs is observed in less than 0.01% of the total GISAID EpiCoV genomes.**

These thresholds were selected according to previous studies (<https://academic.oup.com/ve/article/7/1/veab013/6137845>,

<https://pubmed.ncbi.nlm.nih.gov/33602693/>, <https://elifesciences.org/articles/84384>). The reason for (1), (2), and (3) is that we want our method to be conservative and reduce the number of false positive cryptic mutation calls caused by sequencing errors. A minimum AF of 0.02 (or lower) for SNVs is widely accepted for SARS-CoV-2 variant calling (<https://journals.plos.org/plospathogens/article?id=10.1371/journal.ppat.1009499>, <https://www.tandfonline.com/doi/full/10.1080/21505594.2021.1911477>, <https://journals.sagepub.com/doi/full/10.1177/11769343211014167>), as well as minimum read count of 5 (<https://www.mdpi.com/1999-4915/13/1/133>, <https://academic.oup.com/nar/article/50/3/1551/6511974>). On top of that, applying thresholds (2), (3), and (4) ensure that our cryptic lineage identification is conservative and suitable for both low (minimum read count > 5) and high coverage samples (AF > 1%). As a previous studies suggested, using replicate sequencing (in our case, support by multiple samples) with a minimum AF of 0.01 is the recommended combination to reduce the false discovery rate (FDR) while maintaining a low false non-discovery rate (FNR) during variant calling (<https://journals.asm.org/doi/full/10.1128/mbio.01046-23>). The reason for (5) is that we are more interested in novel/rare combinations of SNVs, and want to exclude combinations commonly found in GISAID.

The FDR of our method is affected by multiple factors, including the number of SNVs in the CRs, the AF of the SNVs, sequencing quality, RNA degradation, etc. and is hard to estimate. Indeed, Crykey's FDR heavily depends on the FDR of the variant caller being used, in our case lofreq, which has a FDR of 0.00005% (<https://www.ncbi.nlm.nih.gov/pmc/articles/PMC3526318/>; see section Robustness and false-positive rates).

The following text has been added to the manuscript in accordance with the reviewers' feedback.

> (Methods, Classifying Candidate CRs from Houston Wastewater, Page 9)

The use of multiple filters on the candidate CRs promotes the CR identification process to be conservative. The selection of thresholds and filters were informed by prior studies.

The publications mentioned above are cited in the revised manuscript.

R1Q2. Perform validation of cryptic mutations using long-read sequencing, e.g. Nanopore, especially for CRs with low coverage depth.

We validated a subset of the CRs with PacBio long-read datasets (see Supplementary Figure 1).

Using long-read datasets for the validation of CRs can be challenging. (1) Publicly available long-read sequencing datasets for clinical surveillance do not provide sufficient coverage for . (2) Compared to short-read sequencing platforms such as Illumina, ONT has a much higher error rate and is not suitable for variant calling for minor variants with low allele frequency, especially when the sequencing depth is insufficient. Therefore, we limited our validation to clinical datasets sequenced with high-accuracy Illumina and PacBio HiFi platforms.

R1Q3. What is the origin and timing of these CRs?

The emergence of the CRs coincided with the spread of new variants of concern. We observed an increased number of CRs in Houston wastewater during the emergence of Delta (July 2021) and Omicron (December 2021). As mentioned in the discussion, most CRs did not persist for more than 4 weeks. However, we noted a few exceptions, and these CRs were the focus of our study.

The origin of the CRs remains an open question. We consider five alternative explanations in the Discussion: (1) under-sampling in clinical surveillance, (2) intra-host low-frequency mutations in the population, (3) signal from SARS-CoV-2 transcription such as subgenomic mRNAs, (4) non-human hosts, and (5) technical artifacts from various sources.

The following text has been added to the manuscript in accordance with the reviewer's feedback.

>(Discussion, Page 5)

The emergence of the CRs coincided with the spread of new variants of concern. We observed an increased number of CRs being detected in Houston wastewater during the emergence of Delta (July 2021) and Omicron (December 2021).

R1Q4. Does the persistent detection of CRs in wastewater reflect the community surveillance data or is it due to immunocompromised individuals shedding these viruses with CRs?

The characterization of persistent CRs requires additional research. If immunocompromised individuals are responsible for persistent CRs, they should be somehow numerous, which is not easy to envision, and their persistent CRs should be restricted to particular wastewater stations, which is not the case.

R1Q5. Compare and contrast Crykey with previous methods in terms of speed, accuracy, sensitivity, etc.

To the best of our knowledge, Crykey is the only computational method to detect cryptic co-occurring mutations from SARS-CoV-2 wastewater surveillance at a whole genome-scale, so sensible comparisons are not possible. The closest existing method is COJAC (<https://www.nature.com/articles/s41564-022-01185-x>), which searches for co-occurring mutations in BAM files. However, COJAC does not provide any information about whether the co-occurring mutations are novel or rare (cryptic) compared to public records, the focus of our tool. Therefore, Crykey is unique concerning end-to-end cryptic lineage detection.

Nevertheless, Crykey is built based on widely used published tools (Iofreq, BWA-MEM, samtools, VDB, etc) that have been previously benchmarked individually.

R1Q6. In Introduction, after identifying what is unknown, explicitly state the objectives of the study.

We agree that the objectives of the study are important and should be listed out clearly in the introduction section. We have added the following paragraph.

>(Introduction, Page 2)

The objective of this study was to (1) develop a tool that enables the detection in wastewater samples of cryptic lineages that have not, or rarely been reported in GISAID's EpiCoV database, (2) investigate the possible origins of these cryptic lineages by contextualizing wastewater and clinical surveillance data.

R1Q7. In Methods, why is the study period of clinical data short (8 weeks) vs. 3 years for wastewater?

The 8-week period was largely due to the fact that we relied on high-quality clinical data specific to Houston, TX. The only publicly available clinical dataset we found was limited to samples collected over 8 weeks.

R1Q8. In Results, show a figure that parallels the workflow, which tracks the initial number of sequences down to identifying CRs.

We updated Figure 1b to include the wastewater sample count, candidate CR count, and CR count after filtering. The figure legend has also been updated.

Figure 1. Workflow and algorithms of Crykey. a) Crykey constructs a genome-to-mutation database and mutation lookup tables using GISAID sequences and metadata. **b)** Crykey searches for two or more mutations on the same read or read-pair and uses the mutation lookup tables to identify whether the linked read mutations represent a candidate CR. Then, each candidate CR is queried against the genome-to-mutation database to calculate its prevalence rate; if they meet the indicated thresholds, they are then considered a CR. **The number of wastewater samples, candidate CRs, and CRs used in this study after filtering are shown in parentheses.** **c)** Algorithm to search candidate CRs, with an example of a read-pair containing mutations A and B. **d)** Algorithm for the fast, exact search for prevalence calculation, with an example of a candidate CR containing mutations A and B.

R1Q9. In Discussion, restructure the discussion such that it will address the objectives of the study in the early paragraphs.

Multiple edits have been added based on the reviewer's suggestion, which include the following edits (listed below), additional edits associated with R1Q3 in the Discussion, and additional edits associated with R1Q6 in the Introduction.

>(Discussion, Page 5)

Furthermore, most cryptic lineage detection methods require ultra-deep sequencing or combining data from both long and short reads and can not be applied with commonly used wastewater surveillance protocols.

>(Discussion, Page 5)

Our results suggest that the number of detected CRs in wastewater relates to the shift of dominant VOCs in the region.

We thank the reviewer again for the feedback.

Reviewer #2 (Remarks to the Author):

This is a well-written and clear paper that presents a novel tool that would allow the identification of and determination of the frequency of cryptic lineages among sequencing samples. The paper is significant in that it presents a solution to known problems associated with these types of variants and pays attention to the current limitations and challenges associated with the work. The tool has broad applicability to viruses and potentially to other small genomes owing to the design of the tools in that it takes into account bins of time to sample from.

R2Q1. A possible addition would be to restrict the sampling to geographic locations owing to the geographic distributions of emerging viral variants?

We thank the reviewer for the constructive suggestions. We are not sure we completely understand this point. We do not have a geographic distribution of emerging viral variants in Houston from clinical samples to correlate with CR emergence. We also looked into the geographic distribution of the CRs in different wastewater treatment plants around Houston, but did not find correlation with CR emergence as well.

I am not aware of another program that looks at the whole genome and reports specifically on these types of mutation frequencies given the definition of cryptic mutations that they describe.

To our best knowledge, Crykey is the first computational method to detect cryptic co-occurring mutations from genome-scale SARS-CoV-2 wastewater surveillance.

Specific edits:

We appreciate the detailed edits the reviewer suggested, and changes have been made accordingly.

Line 141 need a space between Figure and 7

> (Results, Page 4)

In contrast, CR2 was detected in the wastewater only during the first two weeks of the sampling period, while also detected with very low prevalence in the clinical samples during weeks 1-6 (Figure 7).

Line 144 are found in "A" much lower number...

> (Results, Page 4)

As expected, the consensus level mutations are often found in millions of SARS-CoV-2 sequences, and the mutations with low AF are found in a much lower number of sequences...

Lines 149-164 the tenses throughout this paragraph are confusing - I'd prefer all current or past tense...

Line 158 reaching "a" peak

Line 164 than "in" clinical samples

> (Results, Page 4)

As the prevalence rate in clinical samples increased, we could detect CR8 in wastewater on the 3rd week (Figure 8), from samples sequenced using distinct protocols. CR8 consisted of two mutations, C10449A and T10459C. C10449A was a consensus-level mutation for Omicron strains, and it had an individual mean AF close to 1 in both wastewater and clinical samples (Figure 8a and Figure 8e). The prevalence rate of C10449A alone gradually increased in the first 3 weeks of detections starting from the week of 2021-12-06, until the prevalence rate reached 1, and the pattern was consistent in both wastewater and clinical samples; on the other hand, mutation T10459C was present as a low-frequency mutation with individual mean AF close to 0.02. The prevalence rate of T10459C alone in clinical samples increased in the first half of the sampling period, reached a peak at week 4, and then decreased in the second half of the sampling period (Figure 8a and Figure 8e). Since CR8 contained both a consensus level mutation C10449A and a low-frequency mutation T10459C, both mean AF and prevalence rate of the co-occurring mutations followed the pattern of the T10459C (Figure 8b and Figure 8f), and as the prevalence rate of CR8 in clinical samples increased, we started to detect it in wastewater on week 3 as well. The average number of reads that span CR8 regions are shown in Figure 8c and Figure 8g as coverage and Crykey is sensitive enough to detect CR8 in wastewater, given that the coverage of wastewater samples was much lower than in clinical samples (Figure 8d).

Line 174 Maryland had "a" much higher proportion

> (Results, Page 5)

...and Maryland had a much higher proportion of BA.1.18 and BA.1.20 as well.

Line 178 fix the reference

> (Discussion, Page 5)

Wastewater monitoring for SARS-CoV-2 has been widely used for complementing clinical genomic surveillance during the COVID-19 pandemic^{14,31}.

Line 259 and "testing" multiple read aligners

> (Discussion, Page 7)

...and testing multiple read aligners...

Line 295 extra space between cryptic and lineage

> (Discussion, Page 7)

We are hopeful that our findings will help promote community-wide discussion on best practices for cryptic lineage tracking in wastewater.

Line 298 occurring "in" less than

Line 299 extra space between) and in wastewater

Line 297-300 consider a rewrite of this whole sentence, it is confusing as written

> (Methods, Page 8)

Crykey is a computational method for identifying cryptic mutations representing potential cryptic lineages (CRs) in wastewater samples on a full-genome scale. We identify cryptic mutations as sets of two or more co-occurring mutations in the same sequencing read that appear in 5 or more reads, with a minimum AF of 0.01 in two or more samples and observed together in less than 0.01% of the GISAID EpiCoV genomes (up until 10/21/2022).

Line 305 Then, "each"

> (Methods, Page 8)

Then, each CR candidate is queried against a pre-built database to check if it is novel or rare in terms of prevalence to create candidate CRs (Figure 1d).

I don't believe that additional work is needed to demonstrate the conclusions presented and the claims herein. The authors have endeavored to present all possible interpretations and have honestly considered the limitations. I do not detect any flaws in the data analysis, or the interpretation or conclusions. The methodology appears sound. I believe it meets standards.

Please accept our greatest appreciation. We are flattered by the reviewers' positive view of our manuscript.

The detail and instructions provided are likely sufficient to support testing and implementation of the method. As with many other algorithms and tools, the use of the tool will likely reveal some needed modifications in future derivations.

REVIEWERS' COMMENTS

Reviewer #1 (Remarks to the Author):

The issues raised were satisfactorily answered by the authors. I have no more comments.

Reviewer #2 (Remarks to the Author):

The authors have considered all my concerns